# Perspectives on Antibiotic Stewardship Programs among Health Care Providers at Two University Hospitals in Egypt

**DOI:** 10.3390/ijerph20053777

**Published:** 2023-02-21

**Authors:** Marwa Rashad Salem, Meray Rene Labib Youssef, Silvia Farouk Shalaby, Ahmed Taher Mahmoud, Mohamed Ismail, Sally Kamal Ibrahim

**Affiliations:** 1Department of Public Health and Community Medicine, Faculty of Medicine, Cairo University, Manial, Cairo 11559, Egypt; 2Department of Pediatrics, Faculty of Medicine, Cairo University, Manial, Cairo 11559, Egypt; 3Department of Critical Care Medicine, Faculty of Medicine, Cairo University, Manial, Cairo 11559, Egypt

**Keywords:** antibiotic stewardship, physicians, university hospital, barriers, qualitative

## Abstract

The perspectives of healthcare professionals on antibiotic stewardship programs (ASPs) should be explored. Any antibiotic stewardship strategy must be individualized based on patient needs, prescription habits, and local resources. The current study aimed to explore the perspectives of healthcare providers on antibiotics stewardship and their awareness of these perspectives. Furthermore, potential barriers to the application of ASPs should be identified and addressed. This exploratory cross-sectional study utilized a qualitative method to evaluate critical care physicians, pediatricians, and clinical pharmacists (*n* = 43). The mean age of the physicians was 32 ± 1.5 years. Among them, approximately two-thirds (66%) were women. A thematic content analysis was performed to examine the responses of the participants and to prioritize the recommendations for and barriers to the implementation of ASPs from the perspective of healthcare providers. According to the interviewees, the primary obstacles include lack of time in implementation and monitoring and lack of awareness of the need for ASPs. All respondents recommended the implementation of supervised and continuous trainings. In conclusion, the abovementioned barriers must be adequately addressed to facilitate the implementation of ASPs.

## 1. Introduction

Antibiotics resistance (AR) is one of the most important issues affecting critical care. The misuse of antibiotics in society and even in hospitals and the increased utilization of broad-spectrum antibiotics have been contributing to the increased incidence of antibiotic resistance [1]. These factors led to a higher incidence of antibiotic-resistant infections [2]. According to the Political Declaration of the High-Level Meeting of the General Assembly on AR conducted on 9 September 2017, and the Global Action Plan on AR at the World Health Assembly conducted on May 2015, AR is an international public health concern [3].

Stewardship refers to the cautious and conscientious management of something entrusted to one’s care [4]. Antibiotics stewardship (ASP) was first used in hospital settings to optimize antibiotic use [5]. Stewardship has been applied in the context of medical sector governance, thereby assuming responsibility for the population’s health and well-being and guiding health systems at the national and worldwide levels [6].

ASP has become an unreplaceable initiative particularly in the critical care units. The first goal of the program is to collaborate with health care providers in treating patients with the most effective antibiotics at the right dose and appropriate timing. This was based on the four Ds of optimum antibiotics treatment by Joseph and Rodvold, which are as follows: right drug, right dose, de-escalation to pathogen-directed therapy, and right duration of therapy [7].

Rationalizing the use of antibiotics is important in managing infections in an efficient manner, preventing complications caused by unnecessary antibiotic treatment, and combating antibiotic resistance. Further, by enhancing antibiotic prescription, ASPs can help clinicians in enhancing clinical outcomes while minimizing risks [8].

According to a large and expanding body of research, hospital-based ASPs can decrease antibiotic resistance, improve treatment outcomes, and facilitate cost efficiency. The practicality of ASPs and the improvement of patient outcomes from such attempts are supported by evidence from different inpatient institutions, which include multiple types of hospitals and long-term care facilities with a broad range of staffing or financial resources. Previous studies have shown the importance of professional training, dissemination of information about the prescription rates between physicians, and resources for selecting the best clinical options in the outpatient context [9].

Thus, the perspectives of healthcare professionals on ASP should be identified. ASPs must be individualized based on patient needs, prescription habits, obstacles, and local resources [10]. A recent study conducted among a group of physicians in Iran revealed that our physicians’ level of knowledge about antimicrobial stewardship is poor [11]. The current study aimed to understand the perspectives of healthcare providers on ASP and to assess their awareness regarding such perspectives. Furthermore, to successfully facilitate ASPs, potential barriers that should be addressed initially were identified.

## 2. Materials and Methods

### 2.1. Study Design and Setting

This was an exploratory cross-sectional study that utilized a qualitative method. Critical care physicians, pediatricians, and clinical pharmacists who were working in critical care units in two university hospitals (El Manial Hospital and Children University Hospitals, which are affiliated with Cairo University) were included in the analysis. Then, their opinions and perspectives on the implementation of ASP programs were evaluated. The current study was performed in accordance with the CORE-Q unified criteria for reporting qualitative research [11].

### 2.2. Sample Size and Sampling Technique

Participants were enrolled using the purposive sampling technique. The eligibility criteria were as follows: female and male physicians or pharmacists who worked in the critical care units and who provided consent.

When no novel themes, subthemes, or explanations arose from the interviews, data saturation was reached [12]. In total, 50 physicians and pharmacists were invited to join the study. However, we only received feedback from 43 pharmacists and physicians who agreed to participate in a 45 min interview between May and July 2022.

### 2.3. Data Collection

In-depth interviews were performed by team members with experience in qualitative interviews and knowledge on stewardship. All interviews were conducted by one investigator. Two investigators ensured that the content and interview approach were consistent. The interviews were audio recorded. In relation to this, an interview guide was created based on domains identified via a literature review [13,14,15,16,17,18], as shown in Table 1.

### 2.4. Data Analysis

The researchers used a collaborative codebook generation and coding approach with several steps [19]. To facilitate the systematic coding process of discovering themes from qualitative data using the thematic analysis method, transcripts were loaded into the NVivo software version 10.0 (QSR International, Burlington, MA, USA).

To identify the key themes arising from the data, the interview transcripts were coded, improved, and categorized. Initially, the transcripts were broadly coded to create a preliminary coding system. Thereafter, additional codes were created if new themes evolved. Emerging topics and subthemes were regularly polished, tested, and altered using the constant comparison technique. Data were analyzed until all themes were covered. Existing themes were refined and validated via discussions with co-researchers until a consensus decision was achieved. The researchers used Excel 2010 to store demographic data. The quantitative variables were expressed as the mean and standard deviation and qualitative variables as numbers and percentages.

### 2.5. Ethical Considerations

The research protocol was approved by the Ethics Review Committee of the Faculty of Medicine Cairo University (Approval number = N-141-2022). The study was performed according to the Declaration of Helsinki. The participants were informed about the aims of the study, and each of them provided a written informed consent. The identity of the informants remained confidential throughout the study. “I” (interviewee) accompanied by a number was used for each participant based on the chronological order of the interviews (I01, I02, I03 … I13).

## 3. Results and Discussion

### 3.1. Characteristics of the Participants

As the datasets had reached saturation, 43 interviews were performed, without new discoveries arising from the interviews. The HCPs interviewed 19 pediatric physicians, 10 critical care physicians, 10 pharmacists, and 4 clinical pathology physicians. The average age of the physicians (*n* = 43) was 32 ± 1.5 years. Among them, approximately two-thirds (66%) were women. The ASP-related experience of participants varied significantly, ranging from zero to three years. Meanwhile, their average weekly ASP activity time ranged from 0 to 30 h. Only three physicians and two pharmacists received ASP training.

Thematic analysis of interview transcripts revealed the following themes, which indicated the challenges faced when establishing ASP and the approaches that participants believed were required (either innovative techniques or revisions of the existing programs).

### 3.2. Barriers

#### 3.2.1. Time Constraints

When asked about specific team members and time restrictions, the respondents believed that they or other pharmacists and physicians who are core members of the team do not have adequate time for antibiotic stewardship. Part-time allocation was common among the responders, particularly physicians, as they had difficulties in allocating time for stewardship.

One responder believed that time limitations and personnel concerns were significant obstacles to the implementation of an effective program and that the impact of the program was constrained by the incapability of performing certain duties, which involve the proper supervision of health care providers. When interviewed about AMPs, the professionals stated that they could not engage in this program even though it is beneficial. This finding is consistent with that of a study by Mathew et al. [20]. That is, one of the participants stated that implementing stewardship programs can be challenging, particularly in crowded outpatient settings.

#### 3.2.2. Lack of Awareness on ASP

The application of ASP was limited by the lack of awareness. Most participants at two of three sites were not aware of the ASP at their hospitals or the availability of resources that can be used to find this information. However, six respondents from one facility were aware of the ASP. Lack of awareness on ASPs among health care personnel inhibited the application of functional and efficient stewardship. Similarly, some misunderstandings on the importance of selective reporting about AST findings and inconsistent information on its usefulness, efficacy, or application limited its implementation (as stated by the majority of participants).


*There is a lack of local antimicrobial regulations and procedures in orientation programs for new health care workers, which has contributed to the frequently incorrect use of antibiotics. One physician stated.*


According to a study by Shallcross et al. [21], behavioral change is essential in the combat against antibiotic resistance. AMS programs can encourage and monitor the prudent use of antibiotics. However, behavioral and social variables are rarely considered when providing treatments. Hence, the development of a research program aiming at co-designing AMS interventions across healthcare settings by merging data science, proof synthesis, behavioral science, and user-centered design was proposed [18].

#### 3.2.3. Monitoring of ASP

Regarding the impact of programs, most participants responded that continuous monitoring with feedback ASP statistics, preferably via direct communication, subsequently led to the continuous support from local leadership and expanded involvement via the organization. The result of the current study was in contrast to that of a recent study conducted by Monmaturapoj et al. [22].

#### 3.2.4. ASP Guidelines

Most participants at one of the three sites stated that there are no available AS stewardship guidelines. Moreover, the other participants stated that the *lack of teamwork and guidance* is one of the challenges that limit the application of AMPs. This finding is in accordance with that of a study conducted by Mathew et al. [20] who showed that a sound management framework is required for implementing initiatives such as stewardship. No one, in the junior resident’s perspective, is responsible for medicines that are destined for a pharmacy [18].

## 4. Recommendations

### 4.1. Healthcare Professional Training

The participants reported that education and training are important factors in the implementation and adoption of ASPs. Workshops for improving awareness on AR, education and training on antibiotics policies and guidelines, and appropriate ASP should be included. The participants, particularly senior physicians, emphasized the importance of increased awareness on AMR and education on the proper use of antibiotics.

One pharmacist said, “This is really an issue because even the frontline personnel don’t know exactly what we need to do”.

All participants agreed that training is required before ASPs can be implemented. One physician stated that “Yes. Training in the form of workshops is essential to obtain a deeper understanding, and I feel that healthcare providers should have certifications from an accepted course before they can be a part of the ASP”. This finding was consistent with that of the study conducted by Charani et al. [23], who emphasized the importance of developing long-term and efficient ASP training programs for pharmacists and nurses.

#### 4.1.1. Supervision

Only a minority of interviewees were aware of ASPs, and the majority agreed that executing a program with clearly defined goals and objectives and precise definitions of the role of key stakeholders is necessary. Moreover, the use of antibiotics should be audited, monitored, and managed similar to financial resources. All physicians, except five, agreed that combined supervision and antibiotic limitations could be significantly beneficial. “*Audit is a really good intervention. However, it is resource-intensive and may be challenging to implement in all wards*”. One participant said that *“Audits must be performed in certain wards*”.

#### 4.1.2. Resources

Human and non-human resources are important to effectively facilitate ASPs. The introduction of ASP has run into several difficulties, including lack of fund and information technology to delivering viable and effective ASP. Stated by the majority, this finding was in accordance with that of a recent study. Moreover, it is consistent with the findings of Hayat et al. [24], which showed that the lack of standard diagnostic tools resulted in the inappropriate prescription of antibiotics. According to Black et al. [25], the lack of resources limits the increased use of antibiotics.

## 5. Conclusions

According to the interviewees, the primary obstacles include lack of time in implementation and monitoring and lack of awareness on the need for ASPs. All respondents recommended the implementation of supervised and continuous trainings. The abovementioned barriers must be adequately addressed to facilitate the implementation of ASPs.

## 6. Limitations

The current study had several limitations. That is, it only included personnel from two hospitals affiliated with one university hospital. As a qualitative method for adopting antibiotic stewardship programs at critical care units that is not designed to generalize the phenomenon, such limitations did not affect the results. However, multicenter studies utilizing different research techniques should be conducted in the future.

## Figures and Tables

**Table 1 ijerph-20-03777-t001:** Interview Guide.

Background characteristics	Age, sex, title, institution Years of experience in your specialty Role in the organization Professional education
Perspectives	Did you receive any training about antibiotic stewardship?
	Are you involved in implementing the antibiotic stewardship program? If so, what is your role?
	Barriers to the implementation of antibiotic stewardship.
	How could these barriers or challenges be addressed?
	What are the benefits of implementing the antibiotic stewardship program?
	To what extent does ongoing training appropriately address the learning requirements of healthcare workers? Are there any gaps?
	What additional endorsement or resources are needed to enhance antibiotic stewardship programs?
	To what extent do you think healthcare workers are in general compliant with these guidelines?
	Does your organization have guidelines for antibiotic stewardship programs?
	Who is responsible for monitoring antibiotic stewardship programs within your organization?
	What are your current antibiotic stewardship practices and your role in antibiotic stewardship programs?
	Your thoughts on the precise stewardship initiatives that must be prioritized and the ideal methods for including healthcare professionals in stewardship initiatives are constrained.
	Any recommendations?

## Data Availability

Not applicable.

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
