# Peer review of "Perspectives on Antibiotic Stewardship Programs among Health Care Providers at Two University Hospitals in Egypt"

_ijerph, 2023, doi:10.3390/ijerph20053777_

Round 1
Reviewer 1 Report
Overall this is important information to understand ASP knowledge , awareness, and practice in middle income countries, especially in urban hospital settings. Below please find suggestions to make the paper stronger through clarification and contextualization.
Abstract need editing for clarity- for example line 30 should read “the average [age] of physicians…” Should include the physician specialty.
Intro-
Line 43- define postponed de-escalation techniques in the context of stewardship strategies when you first use the term.
Line 73- specify critical are healthcare professionals since that is your target pop. Would be beneficial to contextualize this study within the literature on ASP in lower-middle income countries-i.e. is there a lot of data on this or does it mostly focus on high income countries? If there is literature, what does it say?
Need to expand on the current literature about knowledge attitude and practice of ASP in critical care to provide context and need for your study.
Materials and Methods-
2.1. Why were these two hospitals chosen? Do they have an ASP program in place? Hospital size and population served? These are important contexts for understanding your study data.
2.2 How were subjects recruited? You say purposive sampling focused on critical care but in section 3.1 you say the data included pediatricians and clinical pathologists. Were they involved in critical care?
2.3 Line 73 “performed by a team member” since there was only one interviewer. How did the other two team members ensure that the content was consistent ? development of the interview guide? When you asked “ To what extent do you think healthcare workers are in general compliant with these guidelines?” If they were not familiar with the guidelines did you provide them during the interview?
2.4 So inductive coding was used? Please clarify. There were three coders?
Results –
I would recommend a separate discussion section as the reporting of results and discussion in the same section makes it confusing for the reader what themes and information was found in your study
3.1 So physicians and pharmacists were trained in certain sub-specialties but all involved in critical care? If not how were they chosen in purposive sampling? Please clarify in 2.2 and in this section.
Barriers
4.1. Line 145, so the physicians were part-time at the hospital? Line 153-I thought they were critical care physicians? Do they also do out patient work? Was their perception different in-hospital and in out patient settings?
4.2- Is there an ASP program at these two hospitals that could be aware of?
4.3-I thought that only two had exposure to ASP programing? Is this feedback from those two?
In general you should discuss if there were differences by specialty, length of time practicing, etc… for the reported outcomes.
Contextualizing the information within a comparison of other middle income country settings and/or with high income settings would make this paper a useful starting point for other hospital practitioners.
Conclusion-
Limitations line 227-You indicated it was from two university hospitals-this says one-which is correct?
Author Response
Thanks a lot for your valuable comments

Reviewer 2 Report
This is a timely and important topic. The qualitative study is well-designed and the conclusions seem valid, but without including more quotes from the study participants the paper seems more like an opinion piece. Authors should include multiple quotes (ideally 3) for each key point they are making - this will increase both the understandability of and enthusiasm for, this manuscript.
Author Response
Many thanks for your kind comments.
The changes are highlighted in blue.
Regards

Reviewer 3 Report
See attached

Author Response
Greetings,
Kindly find the attached reply file
Regards,
Marwa

Round 2
Reviewer 1 Report
Addressed the majority of concerns and clarified key issues.
Reviewer 2 Report
The changes made by the authors in this submission, compared to the first, increase the overall readability. The inclusion of more direct quotes gives more weight to the qualitative nature of the paper and helps illustrate the main discussion points raised by the authors.
Reviewer 3 Report
None